ecology/environmental science

atolls, habitat loss, human land use, Small Island states, invertebrate decline, remote sensing

**Author for correspondence:**
Christian Laforsch
e-mail: christian.laforsch@uni-bayreuth.de

# Tourism and urban development as drivers for invertebrate diversity loss on tropical islands

## Sebastian Steibl[1], Jonas Franke[2] and Christian Laforsch[1]

[1]Department of Animal Ecology I and BayCEER, University of Bayreuth, Universitaetsstr. 30, D-95440 Bayreuth, Germany
[2]Remote Sensing Solutions (RSS) GmbH, Dingolfingerstr. 9, D-81673 Munich, Germany

iD SS, 0000-0003-4819-8581; CL, 0000-0002-5889-4647

Oceanic islands harbour a disproportionately high number of endemic and threatened species. Rapidly growing human populations and tourism are posing an increasing threat to island biota, yet the ecological consequences of these human land uses on small oceanic island systems have not been quantified. Here, we investigated and compared the impact of tourism and urban island development on ground-associated invertebrate biodiversity and habitat composition on oceanic islands. To disentangle tourism and urban land uses, we investigated Indo-Pacific atoll islands, which either exhibit only tourism or urban development, or remain uninhabited. Within the investigated system, we show that species richness, abundance and Shannon diversity of the investigated invertebrate community are significantly decreased under tourism and urban land use, relative to uninhabited islands. Remote-sensing-based spatial data suggest that habitat fragmentation and a reduction in vegetation density are having significant effects on biodiversity on urban islands, whereas land use/cover changes could not be linked to the documented biodiversity loss on tourist islands. This offers the first direct evidence for a major terrestrial invertebrate loss on remote oceanic atoll islands due to different human land uses with yet unforeseeable long-term consequences for the stability and resilience of oceanic island ecosystems.

## 1. Introduction

The growing human population and its increasing land and resource demands have altered the whole biosphere and caused a major impact on ecosystems worldwide [1,2]. Globally, the human-driven loss of ecosystems reduced the average species

abundance by 88% compared with its value before human impact [3] and within the foreseeable future, between 33 and 66% of all species worldwide may probably disappear [4]. However, the rate of biodiversity loss differs markedly between different systems [5]. Oceanic islands exhibit one of the fastest rates in human-driven biodiversity loss [6]. At the same time, they harbour about 20% of all species, and 50% of all endangered species, despite constituting only 2.5% of the Earth's surface [7]. Understanding the human impacts on oceanic island ecosystems is thus of particular significance for monitoring and countering the global biodiversity loss [8].

Biological invasions and resource exploitation have been long regarded as the principal drivers of biodiversity loss on oceanic islands [9,10]. With the growing economic development of most island states, increasing human land demands and land use-related system modifications are predicted to become predominant threats to island biodiversity in the future [7]. Land use on oceanic islands is primarily driven by urban development, due to rapidly growing human populations [11,12]. With limited agricultural land available, tourism often constitutes the biggest and fastest-growing economic sector for island states, thus further driving human land consumption and modifications on oceanic islands [13,14].

Other than urbanization with its large-scale infrastructure and city development, tourism land use includes a series of system modifications, like the development of tourism infrastructure, golf courses, landscape gardening and other leisure-related activities that generally differ from urban developed areas [15–17]. Frequently, tourism land use can overburden local waste-management leading to pollution of surrounding ecosystems [18]. Increasing transportation and leisure activities are leading to direct disturbances for native biota, like nesting birds, in tourism-developed areas [19]. While the environmental impacts of pollution, transportation or individual leisure activities under tourism land use have been demonstrated on mainland and marine systems [19–22], the impact of tourism on oceanic island biodiversity has not been investigated, despite constituting a major and increasing form of land consumption on these island systems ([20,22,23], but see [24]).

In this study, we disentangled the environmental impacts of tourism and urban land use on oceanic islands. We achieved a clear spatial separation of these two different forms of human land use by investigating small atoll islands that either harbour only tourist facilities, or are inhabited by the local communities, or remained completely uninhabited [20]. We focused on the ground-associated invertebrate community, as it commonly forms the most diverse and abundant species group on small oceanic islands and carries out various ecological functions critical for the stability and resilience of island ecosystems [23,25,26]. We combined *in situ* sampling of the ground-associated invertebrate community with island-wide geospatial information derived from very-high-resolution satellite data. We tested for differences in diversity indices and in land use/land cover (LULC) between uninhabited islands as control sites free of permanent human land use, and islands under tourism or urban land use. We then tested if LULC changes (i.e. reduction in overall available habitat area, reduction in vegetation density and increase in habitat fragmentation) on tourist and urban islands can be linked to the observed differences in biodiversity, relative to uninhabited islands.

## 2. Methods

Field sampling was carried out in the Lhaviyani (Faadhippolhu) atoll, Republic of Maldives, from 1 February to 26 March 2019. In total, 12 islands were sampled: four uninhabited islands free of any direct and permanent human land use (uninhabited islands), four resort islands used for international tourism (tourist islands) and four islands permanently inhabited by the local Maldivian population. The inhabited islands had total human populations of *ca* 800–5000. Due to their small total area, this results in extremely high population densities (3000–8000 inhabitants km$^{-2}$), comparable or even exceeding those of many metropolitan continental urban areas. Therefore, these inhabited islands meet the criteria of most common definitions of urban areas and were referred to as 'urban islands' [27] (for a map of the study site see [20]). The average island size of each island type was uninhabited islands: 4.91 ± 4.36 ha, urban islands: 40.31 ± 17.85 ha and tourist islands: 18.38 ± 15.85 ha (mean ± s.d.).

On each island, 20 1 × 1 m plots were distributed over the island area using a grid scheme and randomly picking 20 sampling grids. If a plot was positioned in an inaccessible area (e.g. cemetery on urban island, private guest area on tourist island), the plot position was moved for 2 m until the plot lay in an area where it could be sampled. The position of each plot was marked using GPS (Garmin Ltd, Schaffhausen, Switzerland).

Ground-associated macroinvertebrates in each plot were counted and identified to the lowest possible taxonomic level, i.e. species or genus level. One beetle (*Carabidae* sp.) and three spider taxa (*Chelicerata* sp. 1, *Chelicerata* sp. 2, *Theridiidae* sp.) could not been identified to a lower taxonomic level and were treated as four unidentified morphospecies. Ghost crab (*Ocypode cordimana*) abundance was measured by counting the number of burrows within each plot, a standard procedure for estimating their population size [28].

For the remote sensing-based analysis of the landscape parameters, very-high-resolution SkySat data (Planet Labs Inc., San Francisco, USA) were acquired in April 2020. SkySat acquires data with a spatial resolution of 1 m with four spectral bands (visible red, green, blue, near-infrared). The SkySat data were used for LULC mapping and for assessing the vegetation fraction of the investigated islands. An object-based image classification was applied to the SkySat imagery using eCognition (Trimble, Germany). The classification scheme considered the classes infrastructure, water bungalow/jetty/wavebreakers, bare soil/sand, tree cover, shrub and grass/sparse vegetation. The final LULC statistics per island were generated using ArcMap (ESRI, Redlands, USA), excluding the class 'water bungalow/jetty/wavebreaker', since these features are not located on the islands. To evaluate the accuracy of the LULC map, standard procedures for accuracy assessment were followed [29]. Stratified random sampling, using the land cover classes, was chosen for the sampling of the 396 reference locations. This ensured that a minimum number of observations could be randomly placed in each land cover class, while a minimum distance between reference points of 50 m applied. These reference points were manually categorized by an independent image interpreter who was not involved in the classification task. The SkySat imagery and Google Earth data were used for interpreting LULC. The comparison of the classification to the reference data showed an overall accuracy of 88%. The fragmentation of LULC classes was analysed using the fragmentation tool of SAGA-GIS (SAGA User Group Association, Hamburg, Germany) [24,30]. For each island, the defined LULCs were reclassified into three classes. All classes referring to natural vegetation (tree + shrubs + grass/sparse vegetation) were aggregated to 'vegetation area', all infrastructure were aggregated into the class 'other land cover' and the bare soil/sand class was kept as is. The activity in settlement areas (infrastructure) influence fragmentation and must therefore be incorporated. To assess the fragmentation for both natural habitat types, i.e. vegetated areas and bare soil/sand areas, two fragmentation analyses were conducted separately by changing the primary input class. The fragmentation tool first derives two parameters from the aggregated land covers, a density index and a connectivity index [31]. These indices are then used by the fragmentation tool to create the output, which are five fragmentation classes for each island: 'Core', if density = 100%; 'Perforated', if density greater than 60% and density is greater than connectivity; 'Edge', if density greater than 60% and density is less than connectivity; 'Transitional', if density is between 40 and 60%; 'Patch', if density is less than 40%. The percentage area of the 'Core' areas per island was used as the main parameter to assess the influence of habitat fragmentation on biodiversity.

The vegetation cover fraction per 1 m pixel was derived by a partial unmixing method, the mixture tuned matched filtering (MTMF) using the software ENVI/IDL (Harris Geospatial Solutions, Broomfield, USA). MTMF combines parts of a linear spectral mixing model with a matched filter (MF) model and estimates subpixel target abundance [32,33]. MF assesses the spectral signature of a pixel for good matches to the end-member spectrum while suppressing background spectra. An MF score of 1.0 is a perfect match and background material (unknown end-members) is centred around zero [33,34]. For the MTMF, an end-member was used representing a pure vegetation spectrum that was then applied to all data for estimating the vegetation cover fraction in per cent per 1 m pixel. The statistics (mean and s.d.) of the vegetation cover fraction per island were derived using ArcMap (figure 1).

Statistical analysis was done using R v. 4.0.4 [35], extended with the packages 'vegan' for community data analysis [36], 'hillR' for calculating diversity indices based on the Hill numbers [37], 'lme4' for fitting linear mixed-effect models [31], 'emmeans' to estimate marginal means for generalized linear models [38] and 'lavaan' package for path analysis [39]. Effect of island type on the investigated invertebrate community composition was tested using non-parametric permutational analysis of variance (PERMANOVA), based on a Bray–Curtis-dissimilarity matrix and 4999 permutations. Diversity indices were calculated for each plot using the Hill numbers $q = 0$ and $q = 1$. The first Hill number corresponds to species richness and gives no weight to the relative abundances, whereas $q = 1$ corresponds to the exponential Shannon index and weighs species richness by relative abundance [40]. Hill numbers $q > 1$ were not generated, as plot-wise species richness on tourist and urban islands was often zero, which does not permit calculation of Hill numbers $q > 1$. Total accumulated species abundance was calculated as the sum of all invertebrates per plot. To test for the effect of island type

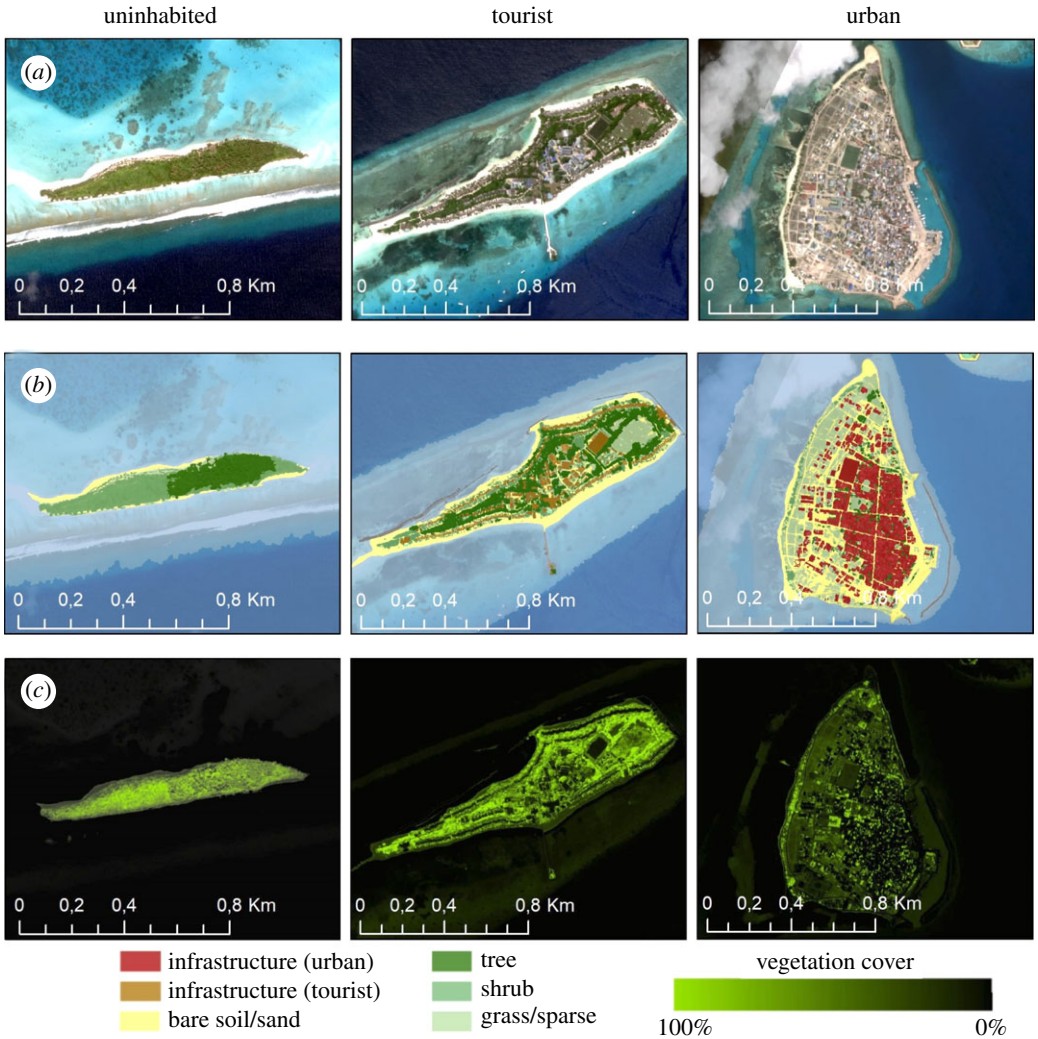

**Figure 1.** Remote-sensing analysis of the landscape parameters. Example images of each island type (uninhabited, tourist and urban) of (*a*) SkySat satellite images, (*b*) land use/land cover classification and (*c*) vegetation cover fraction.

on species richness (Hill number $q = 0$), exponential Shannon diversity (Hill number $q = 1$) and total species abundance, linear regression (ANOVA) using nested generalized linear mixed-effect models with Poisson distribution (for count data) and Gamma distribution (for Shannon diversity) with Tukey HSD *post hoc* testing and 'holm' *p*-value correction were applied. Island size was fitted as a random effect to account for species–area relationships. LULC composition was compared between island types using PERMANOVA. A principal component analysis (PCA) was used as a dimension reduction technique and to account for correlations between different LULCs. The first three principal components (PCs) accounted for 93.54% of the total inertia and were statistically compared between island types using ANOVA and Tukey HSD *post hoc* testing. To generate estimates of the effect of habitat area reduction, vegetation density reduction, fragmentation of the inner insular vegetation and fragmentation of the bare soil/beach habitat on the mean diversity per island, two separate path analysis models were run for the island-wide mean species richness and exponential Shannon diversity (Hill number $q = 0$ and $q = 1$, respectively). The path analysis model showed acceptable fits on the three measures, $\chi^2$ (0.475, d.f. = 2, $p = 0.789$), CFI (1.000) and RMSEA (0.000) for Shannon diversity and taxa richness for the data from the urban islands. However, the model showed no acceptable fits on the three measures, $\chi^2$ (8.141, d.f. = 2, $p = 0.017$), CFI (0.811) and RMSEA (0.620) for Shannon diversity and taxa richness for the data from tourist islands and were thus not further investigated and reported. The overall regression coefficients $R^2$ of the path analysis models were 79.5% and 76.3% for Shannon diversity and richness, respectively. No significant covariances were estimated to occur between the four explanatory variables ($p > 0.05$).

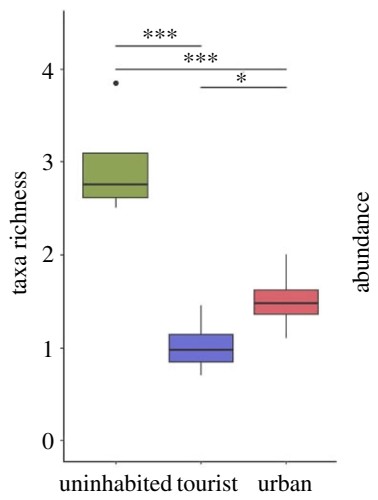

**Figure 2.** Effect of island type on the richness, accumulated abundance and diversity of ground-associated invertebrate communities on tropical islands. Plot-wise taxa richness (*a*), total accumulated species abundance (*b*) and exponential Shannon diversity (*c*) were significantly lower under tourism and urban land uses, compared with uninhabited islands free of permanent human land uses (GLMM: \*\*\**p* < 0.001, \*\**p* < 0.01 and \**p* < 0.05). Plotted for graphical presentation were the mean values for each island (*N* = 4 islands per island type).

## 3. Results

In total, 43 different ground-associated invertebrate taxa were recorded on the investigated uninhabited islands, 31 different taxa on urban islands and 16 different taxa on tourist islands (see electronic supplementary material for an overview of all taxa). The composition of the investigated invertebrate community differed significantly between island types (PERMANOVA: *F* = 3.089, d.f. = 2, *p* = 0.001). Community composition was significantly different on tourist islands, compared with uninhabited (*post hoc*: *p* = 0.003) and urban islands (*p* = 0.022), but community composition did not differ statistically between urban and uninhabited islands (*p* = 0.062).

Island type had a significant effect on the species richness (GLMM: $\chi^2$ = 53.558, d.f. = 2, *p* < 0.001), plot-wise accumulated total species abundance ($\chi^2$ = 16.116, d.f. = 2, *p* < 0.001) and the diversity (exponential Shannon D index) ($\chi^2$ = 66.706, d.f. = 2, *p* < 0.001) of the investigated ground-associated invertebrate communities (figure 2 and table 1). Species richness (Hill number *q* = 0) was significantly smaller on tourist (*post hoc*: *z* = 6.899, *p* < 0.001) and urban islands (*z* = 4.782, *p* < 0.001), compared with uninhabited islands. The negative effect on species richness was thereby larger under tourist land use (Cohen *d* = 1.061) than under urban land use (Cohen *d* = 0.672), relative to uninhabited islands, and species richness further differed significantly between tourist and urban islands (*post hoc*: *z* = −2.341, *p* = 0.019; Cohen *d* = −0.389). Total accumulated species abundance was also significantly smaller on tourist (*z* = 3.546, *p* = 0.001) and urban islands (*z* = 0.724, *p* = 0.002), compared with uninhabited islands. Total accumulated species abundance was not significantly different between urban and tourist islands (*z* = −0.053, *p* = 0.806). Exponential Shannon diversity (Hill number *q* = 1) was significantly smaller on tourist (*z* = −8.078, *p* < 0.001) and urban islands (*z* = −5.082, *p* < 0.001), compared with uninhabited islands. Exponential Shannon diversity was also significantly smaller on tourist islands than on urban islands (*z* = 2.996, *p* = 0.003).

The overall island habitat composition derived from the LULC data differed significantly between the three island types (PERMANOVA: *F* = 2.952, d.f. = 2, *p* = 0.013), yet *post hoc* testing could not identify any significant pairwise differences between island types (*p* > 0.05). Dimensionality reduction using PCA was applied to detect differences and inter-correlations in LULC, fragmentation and vegetation density. The first three PCs explained 93.54% of the total variance in LULC cover (figure 3 and table 2). PC1 is a measure for shrub and tree coverage, as well as vegetation density. Although mean PC1 values for each island type suggest higher shrub and tree cover and overall vegetation cover on uninhabited islands than on tourist and urban islands, PC1 scores were not statistically different between island types (ANOVA: *F* = 0.316, d.f. = 2, *p* = 0.737). PC2 is a measure for bare soil/sand and grass coverage,

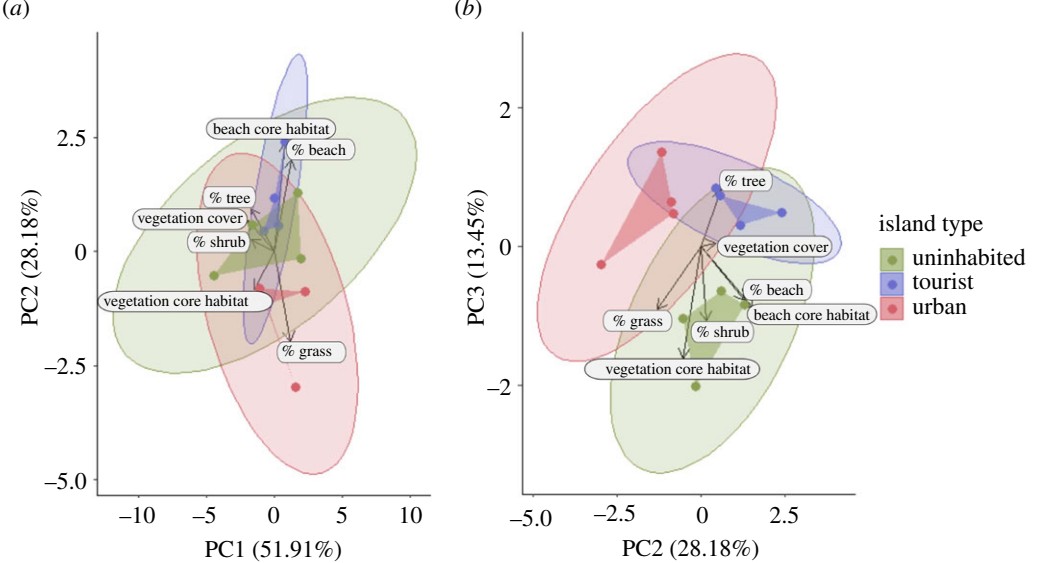

**Figure 3.** PCA of the LULC data and system modifications on the three investigated island types. Each data point denotes the PC values of one island ($N = 4$ islands per island type). The standard ellipses for each island type are assuming multivariate $t$-distributions. For loadings of PCs refer to table 2. (a) PC1 and PC2 bi-plot; (b) PC2 and PC3 bi-plot. All three PCs together explain 93.54% of total variance.

**Table 1.** Diversity metrics and LULC data for the three island types. For the three island types, the mean ± s.d. of the diversity metrics and the relative proportion of each LULC to the total island area is presented ($N = 4$). Core vegetation and core bare soil/sand of the LULC data measure the area of core habitat, with larger values indicating greater connectedness and lower values higher fragmentation of the inner vegetation and bare soil/sand habitat, respectively.

| | island type | | |
|---|---|---|---|
| parameter | uninhabited | tourist | urban |
| richness | 3.0 ± 0.6 | 1.0 ± 0.1 | 1.5 ± 0.2 |
| accumulated abundance | 9.9 ± 3.5 | 6.1 ± 5.2 | 5.9 ± 6.0 |
| exponential Shannon | 2.3 ± 0.4 | 1.2 ± 0.2 | 1.5 ± 0.2 |
| bare soil/sand | 33.9 ± 15.3% | 34.6 ± 6.5% | 22.5 ± 8.3% |
| grass/sparse vegetation | 20.9 ± 14.0% | 8.9 ± 4.2% | 22.5 ± 8.3% |
| shrub vegetation | 25.4 ± 14.8% | 14.3 ± 5.4% | 9.7 ± 6.5% |
| tree vegetation | 19.8 ± 15.0% | 23.9 ± 6.2% | 15.7 ± 12.7% |
| infrastructure | 0% | 18.3 ± 3.9% | 22.4 ± 10.1% |
| core vegetation habitat | 41.6 ± 23.6% | 12.2 ± 5.6% | 20.8 ± 14.0% |
| core bare soil/sand habitat | 17.6 ± 9.8% | 18.7 ± 7.6% | 4.5 ± 2.3% |
| vegetation cover | 37.5 ± 16.9% | 36.2 ± 5.9% | 27.6 ± 13.2% |

as well as fragmentation of the bare soil/sand habitat. PC2 scores differed significantly between island types ($F = 8.453$, d.f. = 2, $p = 0.009$). Urban islands had significantly lower PC2 values than tourist islands ($p = 0.007$) and lower scores than uninhabited islands at the margin of statistical significance ($p = 0.057$), indicating higher grass coverage and lower bare soil/sand coverage with increased fragmentation on urban islands. PC2 scores did not differ between uninhabited and tourist islands ($p = 0.420$). PC3 is a measure of shrub coverage and the fragmentation of the island vegetation. PC3 scores differed significantly between island types ($F = 13.330$, d.f. = 2, $P = 0.002$). Uninhabited islands had significantly lower PC3 scores than tourist ($p = 0.004$) and urban islands ($p = 0.004$), suggesting higher shrub

**Table 2.** PCA of the LULC remote-sensing data for the three investigated island types. The first three PCs explained a total cumulative variance of 93.54% and were chosen for subsequent statistical testing. Loadings of each PC indicate relationship with the given LULC parameter. Italicized values (threshold greater than 0.4 or less than −0.4) suggest a clear positive or negative correlation (e.g. PC2 is positively correlated with sand cover and negatively correlated with grass cover. Higher PC2 values thus correspond to higher sand cover and lower grass cover of an island). Mean ± s.d. of PC1–PC3 score for the three island types are presented below ($N = 4$). Different letters behind values indicate significant differences in the PC between island types (ANOVA: $p < 0.05$); same letters indicate no statistical difference ($p > 0.05$).

| parameter | PC1 | PC2 | PC3 |
|---|---|---|---|
| explained variance | 51.91% | 28.18% | 13.45% |
| sand coverage | 0.322 | *0.508* | −0.300 |
| grass coverage | 0.293 | *−0.498* | −0.352 |
| shrub coverage | *−0.442* | 0.065 | *−0.417* |
| tree coverage | *−0.437* | 0.239 | 0.318 |
| vegetation density | *−0.492* | 0.172 | 0.018 |
| core vegetation habitat | −0.374 | −0.210 | *−0.626* |
| core sand habitat | 0.204 | *0.598* | −0.346 |
| uninhabited islands | −0.61 ± 3.06[A] | 0.30 ± 0.81[AB] | −1.14 ± 0.61[A] |
| tourist islands | 0.08 ± 0.66[A] | 1.15 ± 0.90[A] | 0.59 ± 0.24[B] |
| urban islands | 0.53 ± 1.62[A] | −1.45 ± 1.03[B] | 0.55 ± 0.66[B] |

coverage and less fragmented inner vegetation on uninhabited islands. PC3 scores were not statistically different between tourist and urban islands ($p = 0.995$).

A reduction of available natural habitat could not be correlated to mean invertebrate Shannon diversity ($Z = 1.953$, $p > 0.05$) and mean taxa richness ($Z = 1.961$, $p > 0.05$) on urban islands, but vegetation density reduction ($Z = -2.362$, $p = 0.018$; $Z = -2.315$, $p = 0.021$), fragmentation of the inner vegetation habitat ($Z = -2.279$, $p = 0.023$; $Z = -2.363$, $p = 0.018$) and fragmentation of the bare soil/ beach habitat ($Z = -3.070$, $p = 0.002$; $Z = -2.797$, $p = 0.005$) had a significant negative effect on average diversity and richness on urban islands, respectively. The model for tourist islands was not able to estimate any direct relationship between the diversity indices and the explanatory LULC variables.

## 4. Discussion

We disentangled and directly compared the impact of two predominant human land use forms on oceanic islands, i.e. tourism and urban development, on the ground-associated invertebrate communities. On islands with tourism or urban land use, species richness, accumulated abundance and diversity of the investigated island invertebrate community were significantly smaller than on uninhabited islands. Remote-sensing data suggest that habitat fragmentation and the spatial extension of sparse grass vegetation are significantly higher on urban islands, compared with uninhabited islands, and have significant effects on the measured biodiversity, whereas biodiversity loss on islands under tourism land use could not be correlated to LULC data.

Human land uses can drive biodiversity losses by modifying natural habitats and whole ecosystems [41]. Modifications comprise habitat quality loss, habitat fragmentation, loss of natural vegetation cover and density [42,43], and habitat area loss (i.e. a loss in the amount of habitat). As biodiversity is positively correlated with habitat size [44], a loss of suitable natural habitat area is probably a relevant driver for the observed biodiversity loss following both island types as, on average, 18% of the available island area on tourist islands and 22% of urban islands were covered by housing sites and/or infrastructure.

The relative proportion of sparse grassland vegetation was higher on urban islands than on uninhabited and tourist islands. In accordance, a reduction of overall vegetation cover was suggested to have a significant effect on invertebrate biodiversity on islands under urban development. Here, land reclamation and the creation of new construction sites for future settlements are probably the drivers for the overall reduced vegetation cover [45,46]. As vegetation density and cover are positively associated with invertebrate biodiversity, its extensive reduction could be one possible explanation for

the observed biodiversity loss on the islands under urban land use [47–49]. Generalist taxa, e.g. ants, might still find enough suitable ecological niches to persist in this urbanized environment, but ground-associated specialist taxa, which require specific habitat conditions, probably became locally extinct [50–52]. This effect of niche degradation might be even more relevant for the tree- or shrub-associated taxa that entirely lost their habitat due to the removal of tree and shrub vegetation on urban developed islands [53]. In contrast with the urban development on the permanently inhabited islands, tourist facilities are interested in keeping much of the natural forest and shrub vegetation intact to conserve the image of a 'tropical paradise' for their guests [46]. Therefore, a reduced vegetation cover is probably not a relevant driver for biodiversity loss around tourist facilities, underlined by overall similar vegetation cover on tourist and uninhabited islands (36% versus 38%, respectively, table 1).

Fragmentation of the beach and inland vegetation was higher under urban land use than on the other two island types. Further, the fragmentation was estimated to have a significant effect on the measured biodiversity on urban islands. Habitat fragmentation is probably driven by infrastructure development, resulting in a more patchy environment with more habitat edges [42,43]. Besides roads as drivers of fragmentation of the islands' interior, the coastline's ongoing obstruction by harbour sites and coastal defence structures increases the fragmentation of the natural beachline on urban islands [20,46,54]. At a certain point where habitat patches become smaller than a critical threshold, species could become locally extinct [48,49,55]. On tourist islands, where neither large roads nor coastal defence structures are present that are fragmenting the inner vegetated habitat or beaches to a great extent, fragmentation is thus probably not a significant driver for the observed biodiversity loss. Nevertheless, the inner vegetation on tourist islands was more fragmented relative to uninhabited islands. Fragmentation under tourism land use is probably caused by landscape gardening around tourist facilities, where trees and flowers have been planted in a well-planned manner, resulting in small garden patches with small trails in between.

While LULC data could not be linked to the observed losses in ground-associated invertebrate diversity and taxa richness on tourist islands, another process is probably the critical driver for the impoverished invertebrate fauna under tourism land use. All investigated tourist islands applied insecticides around their facilities to diminish fly, mosquito, cockroach and bedbug populations on a weekly to daily basis (2019, personal communication). Therefore, we suggest that the large-scale application of pesticides, including S-bioallethrin, deltamethrin and many other pyrethroid substances as active ingredients (see also PestEx Maldives, Neeolafaru Magu, Male', Republic of Maldives; [18]) probably is responsible for the impoverished invertebrate communities on tourist islands. Although primarily applied to fight high mosquito abundances typical in the tropics [56,57], pyrethroid insecticides like deltamethrin attack the insects' sodium channels and are therefore not specific to single pest species but also impact non-target invertebrates [58,59]. Consequently, insecticide application probably causes a large-scale loss of many ground-associated invertebrate taxa and may therefore be, at least partly, responsible for the observed loss of ground-associated invertebrate taxa under tourism land use.

Remote oceanic islands contribute disproportionately to global biodiversity and harbour a great number of range-restricted and endangered species [4,60]. Due to their small size, these islands experience little agricultural land use, but as for many small island states, the tourist industry is among the biggest and fastest-growing economic sectors and a dominant driver for land consumption [61]. Assessing its ecological impacts is therefore essential to mitigate associated risks for the unique and often endemic flora and fauna of tropical islands around the globe.

Taken together, our findings underline the necessity to disentangle and directly compare different human land uses in order to understand their ecological consequences more comprehensively [20]. We demonstrate that conventional tourism land use and urban development can have severe impacts on the ground-associated invertebrate diversity on remote oceanic atoll islands. Although agriculture is currently considered the predominant driver of the worldwide species decline [62], it is crucial to investigate and consider all human land uses for obtaining a global impact assessment, especially in regions where land use types other than agriculture are predominant. Only by generating a holistic understanding of the different human pressures and their severity on the world's biomes will it be possible to effectively counteract the ongoing global biodiversity loss.

Data accessibility. All data (besides the original Planet satellite data) are stored in a public repository (Dryad) and can be accessed via https://datadryad.org/stash/dataset/doi:10.5061/dryad.kwh70rz31 [63].

Authors' contributions. S.S. and C.L. designed the study. J.F. conducted the remote-sensing analysis. S.S. carried out the fieldwork and data analysis. S.S., J.F. and C.L. wrote the manuscript. All authors gave final approval for publication.

Competing interests. The authors declare no competing interests.

Funding. Financial support for S.S. from 'Studienstiftung des deutschen Volkes' scholarship is gratefully acknowledged.

Acknowledgements. We thank the NGO 'Naifaru Juvenile' and 'Atoll Marine Centre' for providing and organizing infrastructure and accommodation during the fieldwork and the participating resorts and dive centres for allowing us to enter their islands and conducting the sampling. We thank Simon Steiger and Marvin Kiene for their input on the statistical models.

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
