## [Peer Review File · Royal Society Open Science]

Review History

RSOS-210411.R0 (Original submission)

Review form: Reviewer 1

Is the manuscript scientifically sound in its present form?

Yes

Are the interpretations and conclusions justified by the results?

No

Is the language acceptable?

Yes

Do you have any ethical concerns with this paper?

No

Have you any concerns about statistical analyses in this paper?

Yes

Recommendation?

Major revision is needed (please make suggestions in comments)

Comments to the Author(s)

The manuscript investigates the relationship between land use classes and Shannon diversity / species richness of invertebrates along a gradient of anthropogenic impact (uninhabited, local population, touristic use) on 12 islands in the Maldives. The sampling design seems sufficient, and the analysis is satisfactory. The Discussion section is largely speculative in relation to several potential aspects of the study that were never actually assessed. The results are modest, with several non-significant relationships, which is okay but may likely affect impact. The manuscript would benefit for an explicit treatment of island size to control for SAR in path analysis, which should be the basis of results. I recognize they did a separate analysis with island size, but the two should be combined to allow for their interaction. Next, the land cover modeling possesses no validation - a major lacuna which needs to be addressed. This is standard for any RS-based analysis. Finally, the response variable (mean values from all plots) suffers from unnecessary information loss. Why not retain all plots and their variances, so you don't throw out information? Plots by island could then be used for a more robust modeling approach (and thereby informing inference). Some specific row-by-row comments follow:

24-26 - should couch this not among islands in general, but as among islands in your study domain

28 - what does "local" mean here? Need to define at first instance

47 - "will" > "may"

49 - has indicated

80 - is tropical moist forest rare?

88 - again, not clear what "local" means here, I see it defined below (99), but as with acronyms it should be defined at first instance and perhaps avoided in the abstract

124 - need some unbiased measure of accuracy (i.e. validation)

139-145 - recommend converting to table

157-158 - mean of all plots? why not retain all plots and calculate variance, so you don't throw out information. Plot could likewise be used in the modeling.

168-169 - redundant with above

173-174 - are these islands fully (100%) closed-canopy in their natural state? What about natural disturbance regimes?

198 - what does N=4 refer to? Is it not 12 islands? (4 per type?) Please be clear on this.

211-217 - is this not a trivial finding as uninhabited is defined by a lack of these types of covers?

231-232 - this can go in Methods

244-245 - needs reference

248 - but this study does not address cause, only correlation. Please be clear on this.

258-260 - needs reference

296 - this is more speculation that wasn't studied (normally, you'd test a "hypothesis")

533 - 4 islands "per island type", means 12 islands, no?

534-535 - "reduced" is speculation on a historical process which was not explicitly investigated in this study

536 - can these asterisks be moved closer to the box-whiskers?

Review form: Reviewer 2

Is the manuscript scientifically sound in its present form?

Yes

Are the interpretations and conclusions justified by the results?

Yes

Is the language acceptable?

Yes

Do you have any ethical concerns with this paper?

No

Have you any concerns about statistical analyses in this paper?

Yes

Recommendation?

Major revision is needed (please make suggestions in comments)

Comments to the Author(s)

Dear authors,

Thank you for your manuscript. The presented case study comparing invertebrate diversity between islands with different land use is an important contribution to our understanding of human dominated systems. Islands are of special interest as those harbor above average global diversity.

Major aspects:

1) The data contain 240 independent plots (20 x 1 sqm on 12 islands) which are summarized (mean) to 12 values (one per island). This is an unnecessary loss of information that does not adequately use the quality of the data. I would recommend to change the modelling framework to a generalized linear mixed effect model (with Poisson family error as richness values are count data) that analyze the raw plot values. The nested structure (plots within islands) can be included in the random structure of the model. This setting would further allow including information per plot (land cover class, habitat etc) in the analysis, if wanted. It would also allow differentiating an analysis on richness per plot with one on richness per island. It is possible that richness per plot is differently affected by land use than richness per island.

2) I would recommend analyzing differences in similarity in species composition using ordination (e.g. NMDS or CA) on the plot/species data. Possible predictor variables can be fitted with a post hoc fit or directly included in the analysis (CCA). This would allow a visualization of the difference in species composition with land-use types.

3) Islands are special and relevant regarding global biodiversity. They harbor less species per area, when compared to mainland settings, but an above average share of unique, often endemic species. Many of the species driven to extinction by humans have only occurred on islands. This context information and the island setting of this study should be placed in the introduction which currently builds the impression that the chosen islands are a random and representative case study. More background information on the biogeographic setting and land-use context of the islands would be helpful.

Minor aspects:

Title: I would suggest to remove "permanent" and "major". Both words are not needed for the message.

Line 16: Please remove "pristine". These ecosystems have been altered by humans even without permanent settlements.

Line 19: "virtually absent" -> understudied?

Line 17-19: Split in two sentences to improve readability?

Lines 15-21: These sentences are too general for the setting of the case study. You could/should mention that this study is about islands, which are special and above average in their importance for biodiversity.

Line 20: is "urban" the right word for those small settlements?

Line 21: The location of the case study should be included/mentioned here.

Line 48: You should mention the importance of islands for global diversity and their specialists (evolutionary history) here or in lines 77.

Line 78: is "urban" the right word for those small settlements?

Line 83: How do you know that the biogeographic history is the same?

Lines 86-91: this are two predictions derived from the hypothesis that human land use causes the local loss of native invertebrates. Reformulate. How are non-natives treated in the study?

Lines 112-114: You obviously had the expectation that those affect your dependent variable. Why was this not tested and included in the study?

Lines 156-: See my general comment to the statistical analysis.

Map of the study area with two different spatial scales (global context; regional setting).

Figure 2: It is not clear to me if the boxplots represent the values per plot or per island. I would recommend showing the raw values per plot (1sqm). Please indicate significant differences as derived from the PostHoc test with letters. I would recommend to standardize the text size of all labels and reduce the white space by making the figure smaller.

Decision letter (RSOS-210411.R0)

Dear Mr Steibl

On behalf of the Editors, we are pleased to inform you that your Manuscript RSOS-210411 "Tourism and permanent settling as drivers for major invertebrate diversity loss on tropical islands" has been accepted for publication in Royal Society Open Science subject to minor

revision in accordance with the referees' reports. Please find the referees' comments along with any feedback from the Editors below my signature.

Please submit your revised manuscript and required files (see below) no later than 7 days from today's (ie 09-Aug-2021) date. Note: the ScholarOne system will 'lock' if submission of the revision is attempted 7 or more days after the deadline. If you do not think you will be able to meet this deadline please contact the editorial office immediately.

on behalf of Dr Polly Campbell (Associate Editor) and Pete Smith (Subject Editor)
openscience@royalsociety.org

Associate Editor Comments to Author (Dr Polly Campbell):

Comments to the Author:

First, I want to apologize to the authors for the long delay in getting a decision on their manuscript. Both reviewers find value in the work and make useful suggestions for improvement. I note that the point made by both reviewers regarding loss of information when diversity indices are averaged by island rather than using the values from individual plots is similar to one raised during the initial review process. The authors already did a fine job of addressing the first critique in their response to this review but the fact that the issue is raised again suggests that they should explain in the text why the modeling approach suggested by reviewers isn't feasible. I did not notice any suggestions made by the current reviewers that conflict with changes already made following the first review but if these exist the authors can just reference their response to the first review.

Reviewer comments to Author:

Reviewer: 1

Comments to the Author(s)

The manuscript investigates the relationship between land use classes and Shannon diversity / species richness of invertebrates along a gradient of anthropogenic impact (uninhabited, local population, touristic use) on 12 islands in the Maldives. The sampling design seems sufficient, and the analysis is satisfactory. The Discussion section is largely speculative in relation to several

potential aspects of the study that were never actually assessed. The results are modest, with several non-significant relationships, which is okay but may likely affect impact. The manuscript would benefit for an explicit treatment of island size to control for SAR in path analysis, which should be the basis of results. I recognize they did a separate analysis with island size, but the two should be combined to allow for their interaction. Next, the land cover modeling possesses no validation - a major lacuna which needs to be addressed. This is standard for any RS-based analysis. Finally, the response variable (mean values from all plots) suffers from unnecessary information loss. Why not retain all plots and their variances, so you don't throw out information? Plots by island could then be used for a more robust modeling approach (and thereby informing inference). Some specific row-by-row comments follow:

24-26 - should couch this not among islands in general, but as among islands in your study domain

28 - what does "local" mean here? Need to define at first instance

47 - "will" > "may"

49 - has indicated

80 - is tropical moist forest rare?

88 - again, not clear what "local" means here, I see it defined below (99), but as with acronyms it should be defined at first instance and perhaps avoided in the abstract

124 - need some unbiased measure of accuracy (i.e. validation)

139-145 - recommend converting to table

157-158 - mean of all plots? why not retain all plots and calculate variance, so you don't throw out information. Plot could likewise be used in the modeling.

168-169 - redundant with above

173-174 - are these islands fully (100%) closed-canopy in their natural state? What about natural disturbance regimes?

198 - what does N=4 refer to? Is it not 12 islands? (4 per type?) Please be clear on this.

211-217 - is this not a trivial finding as uninhabited is defined by a lack of these types of covers?

231-232 - this can go in Methods

244-245 - needs reference

248 - but this study does not address cause, only correlation. Please be clear on this.

258-260 - needs reference

296 - this is more speculation that wasn't studied (normally, you'd test a "hypothesis")

533 - 4 islands "per island type", means 12 islands, no?

534-535 - "reduced" is speculation on a historical process which was not explicitly investigated in this study

536 - can these asterisks be moved closer to the box-whiskers?

Reviewer: 2

Comments to the Author(s)

Dear authors,

Thank you for your manuscript. The presented case study comparing invertebrate diversity between islands with different land use is an important contribution to our understanding of human dominated systems. Islands are of special interest as those harbor above average global diversity.

Major aspects:

- 1) The data contain 240 independent plots (20 x 1 sqm on 12 islands) which are summarized (mean) to 12 values (one per island). This is an unnecessary loss of information that does not adequately use the quality of the data. I would recommend to change the modelling framework to a generalized linear mixed effect model (with Poisson family error as richness values are count data) that analyze the raw plot values. The nested structure (plots within islands) can be included in the random structure of the model. This setting would further allow including information per plot (land cover class, habitat etc) in the analysis, if wanted. It would also allow differentiating an analysis on richness per plot with one on richness per island. It is possible that richness per plot is differently affected by land use than richness per island.
- 2) I would recommend analyzing differences in similarity in species composition using ordination (e.g. NMDS or CA) on the plot/species data. Possible predictor variables can be fitted with a post hoc fit or directly included in the analysis (CCA). This would allow a visualization of the difference in species composition with land-use types.
- 3) Islands are special and relevant regarding global biodiversity. They harbor less species per area, when compared to mainland settings, but an above average share of unique, often endemic species. Many of the species driven to extinction by humans have only occurred on islands. This context information and the island setting of this study should be placed in the introduction which currently builds the impression that the chosen islands are a random and representative case study. More background information on the biogeographic setting and land-use context of the islands would be helpful.

Minor aspects:

Title: I would suggest to remove "permanent" and "major". Both words are not needed for the message.

Line 16: Please remove "pristine". These ecosystems have been altered by humans even without permanent settlements.

Line 19: "virtually absent" -> understudied?

Line 17-19: Split in two sentences to improve readability?

Lines 15-21: These sentences are too general for the setting of the case study. You could/should mention that this study is about islands, which are special and above average in their importance for biodiversity.

Line 20: is "urban" the right word for those small settlements?

Line 21: The location of the case study should be included/mentioned here.

Line 48: You should mention the importance of islands for global diversity and their specialists (evolutionary history) here or in lines 77.

Line 78: is "urban" the right word for those small settlements?

Line 83: How do you know that the biogeographic history is the same?

Lines 86-91: this are two predictions derived from the hypothesis that human land use causes the local loss of native invertebrates. Reformulate. How are non-natives treated in the study?

Lines 112-114: You obviously had the expectation that those affect your dependent variable. Why was this not tested and included in the study?

Lines 156-: See my general comment to the statistical analysis.

Map of the study area with two different spatial scales (global context; regional setting).

Figure 2: It is not clear to me if the boxplots represent the values per plot or per island. I would recommend showing the raw values per plot (1sqm). Please indicate significant differences as derived from the PostHoc test with letters. I would recommend to standardize the text size of all labels and reduce the white space by making the figure smaller.

===PREPARING YOUR MANUSCRIPT===

===PREPARING YOUR REVISION IN SCHOLARONE===

Author's Response to Decision Letter for (RSOS-210411.R0)

See Appendix A.

Decision letter (RSOS-210411.R1)

Dear Mr Steibl,

I am pleased to inform you that your manuscript entitled "Tourism and urban development as drivers for invertebrate diversity loss on tropical islands" is now accepted for publication in Royal Society Open Science.

on behalf of Dr Polly Campbell (Associate Editor) and Pete Smith (Subject Editor)
openscience@royalsociety.org

Appendix A

Dear Editor

We want to thank you and the two anonymous reviewers for investing their valuable time in assessing our manuscript entitled 'Tourism and permanent settling as drivers for major invertebrate diversity loss on tropical islands' (RSOS-210411) and accepting it for publication after implementing the reviewers' suggestions. In the following we want to address each comment raised by the two reviewers in a point-to-point response.

Reviewer #1

1. "The manuscript investigates the relationship between land use classes and Shannon diversity / species richness of invertebrates along a gradient of anthropogenic impact (uninhabited, local population, touristic use) on 12 islands in the Maldives. The sampling design seems sufficient, and the analysis is satisfactory. The Discussion section is largely speculative in relation to several potential aspects of the study that were never actually assessed. The results are modest, with several non-significant relationships, which is okay but may likely affect impact. The manuscript would benefit for an explicit treatment of island size to control for SAR in path analysis, which should be the basis of results. I recognize they did a separate analysis with island size, but the two should be combined to allow for their interaction."
→ We followed the reviewer's suggestion and included island size as random effect in the Generalized Linear Mixed Effect Model to directly control for the effect of island size on island invertebrate diversity (see response to comment 3 for more details). Revision of the statistical modelling has, however, not changed the outcomes and major findings / implications of our study.
2. "Next, the land cover modelling possesses no validation - a major lacuna which needs to be addressed. This is standard for any RS-based analysis."
→ We have implemented a validation step in the remote sensing data and present the procedure in the revised method section:
"To evaluate the accuracy of the LULC map, standard procedures for accuracy assessment were followed [28]. Stratified random sampling, using the land cover classes, was chosen for the sampling of the 396 reference locations. This ensured that a minimum number of observations could be randomly placed in each land cover class, while a minimum distance between reference points of 50 m applied. These reference points were manually categorized by an independent image interpreter who was not involved in the classification task. The SkySat imagery and Google Earth data was used for interpreting LULC. The comparison of the classification to the reference data showed an overall accuracy of 88%."
3. "Finally, the response variable (mean values from all plots) suffers from unnecessary information loss. Why not retain all plots and their variances, so you don't throw out information? Plots by island could then be used for a more robust modelling approach (and thereby informing inference)."
→ We followed the reviewer's suggestion (see also response to reviewer #2, comment 1) and re-run the statistical analysis using the plot data. We calculated linear regression (ANOVA) using a nested generalized Linear Mixed Effect model with Poisson distribution for the plot-wise count data. We fitted the nested structure of plot-within-island to the model and built three different GLMMs, either with island size fitted as random slope, as random intercept, or without fitting island size at atoll (yet consider that island size is encoded in island, so the term for the nested structure 1|island/plot somehow contains this

information, only as categorical island name instead of the actual numerical value for the island size). We run all three models separately and checked for variance and standard deviations of the random effects. Next, we compared the performances of the three models using corrected AICs (AICc), which are more suitable when sample size is small (in our case: $N = 4$ islands per island type). Model performance did not differ when island size was fitted as slope, intercept, or not treated specifically (yet encoded in the nested term as categorical variable). We repeated the same procedure for the GLMMs for testing the main effect of island type on total accumulated species abundance and Shannon diversity. The actual models can be accessed in the R markdown file deposited alongside the manuscript in Dryad.

4. “Line 24-26: should couch this not among islands in general, but as among islands in your study domain”
 - ➔ We follow the reviewer’s suggestion and have changed the wording so that we specifically refer to the islands investigated in our study:
“. Within the investigated system, we show that species richness, abundance, and Shannon diversity of the ground-associated invertebrate community are significantly decreased on islands used for tourism and on islands with urban development, relative to uninhabited islands”
5. “Line 28 - what does ‘local’ mean here? Need to define at first instance”
 - ➔ We have decided to completely remove the word ‘local’ in the revised version of our manuscript and instead refer to this island type as ‘islands with urban development’, as the term ‘local’ apparently brought confusion to both reviewers. For further explanations and justification on why these islands can be considered as ‘urban developed’ in response to reviewer #2, comment 9.
6. “Line 47 – ‘will’ > ‘may’”
 - ➔ We have changed the wording according to the reviewer’s suggestion (see track changes version of manuscript).
7. “Line 49 - has indicated”
 - ➔ We have changed the wording according to the reviewer’s suggestion (see track changes version of manuscript).
8. “Line 80 - is tropical moist forest rare?”
 - ➔ We have removed the given line completely, following also comment 3 of reviewer #2, to build up the introduction more directly to the insular ecosystem.
9. “Line 88 - again, not clear what ‘local’ means here, I see it defined below (99), but as with acronyms it should be defined at first instance and perhaps avoided in the abstract”
 - ➔ See comment above. We have decided to completely remove the term ‘local’ from the revised version of the manuscript and instead refer to this island type as ‘islands with urban development’
10. “Line 124 - need some unbiased measure of accuracy (i.e., validation)”
 - ➔ We have implemented a validation step in the remote sensing data. See method section in the revised version of the manuscript and response to reviewer #1 comment 2 for details.

11. "Line 139-145 - recommend converting to table"
 - ➔ We decided to keep the bullet-point format, as we think separating this methodological information from the main text complicates instead of facilitates readability.
12. "Line 157-158 - mean of all plots? why not retain all plots and calculate variance, so you don't throw out information. Plot could likewise be used in the modelling."
 - ➔ We followed the reviewer's suggestion and used a Generalized Linear Mixed Effect model with the plot-wise data and island size as random effect. See also response to reviewer #1 comment 3 and method section for more details.
13. "Line 168-169 - redundant with above"
 - ➔ We followed the reviewer's suggestion and removed this redundant information from the method section of the revised version (see track changes).
14. "Line 173-174 - are these islands fully (100%) closed-canopy in their natural state? What about natural disturbance regimes?"
 - ➔ We cannot exclude that any natural disturbance regimes, e.g., tsunamis, might have impacted the vegetation cover on the investigated islands. However, these natural disturbances occur on a much larger spatial scale and should impact all islands within the atoll the same way and at the same time, rather than affecting only one island or island type, thereby confounding our data. Due to their spatial proximity within one atoll, all investigated islands experience the same natural disturbance regimes. Therefore, we can largely exclude that the observed differences in invertebrate biodiversity are due to differences in natural disturbance regimes.
15. "Line 198 - what does N=4 refer to? Is it not 12 islands? (4 per type?) Please be clear on this."
 - ➔ We added the information 'N = 4 per island type', following the reviewer's suggestion.
16. "Line 211-217 - is this not a trivial finding as uninhabited is defined by a lack of these types of covers?"
 - ➔ We agree with reviewer #1 and have thus removed the given line from the results section of the manuscript.
17. "Line 231-232 - this can go in Methods"
 - ➔ We followed the reviewer's suggestion and moved this information in the method section.
18. "Line 244-245 - needs reference"
 - ➔ We added the following citation at the given line: Maxwell et al. (2016) Biodiversity – the ravages of bulldozers, nets and guns. *Nature*, 536(7615): 143-145.
19. "Line 248 - but this study does not address cause, only correlation. Please be clear on this."
 - ➔ We followed the reviewer's suggestion and changed the wording accordingly:
"A reduction of vegetation cover was suggested to be significantly correlated with the biodiversity loss on local islands"
20. "Line 258-260 - needs reference"

→ We added the following citation at the given line: Sánchez-Bayo & Wyckhuys (2019) Worldwide decline of the entomofauna: a review of its drivers. *Biological Conservation*, 232: 8-27.

21. “Line 296 - this is more speculation that wasn't studied (normally, you'd test a ‘hypothesis’)”
→ We agree with reviewer #1 and changed the wording to ‘suggest’.
“Therefore, we suggest that the large-scale application of pesticides [...] likely is responsible for the significant ground-associated invertebrate diversity loss on tourist islands.”
22. “Line 533 - 4 islands ‘per island type’, means 12 islands, no?”
→ Yes. As this formulation seemed to be misleading, we changed it accordingly:
“ $N = 4$ islands per island type”
23. “Line 534-535 – ‘reduced’ is speculation on a historical process which was not explicitly investigated in this study”
→ We agree with reviewer #1 that this formulation was unprecise as we did not investigate temporal changes. Therefore, we changed the wording accordingly (see track changes)
24. “Line 536 - can these asterisks be moved closer to the box-whiskers?”
→ We have completely revised figure 1 following the reviewers’ suggestion.

Reviewer # 2

“Thank you for your manuscript. The presented case study comparing invertebrate diversity between islands with different land use is an important contribution to our understanding of human dominated systems. Islands are of special interest as those harbor above average global diversity.”

→ We thank reviewer #2 for the overall positive feedback and interest in our study.

1. “The data contain 240 independent plots (20 x 1 sqm on 12 islands) which are summarized (mean) to 12 values (one per island). This is an unnecessary loss of information that does not adequately use the quality of the data. I would recommend to change the modelling framework to a generalized linear mixed effect model (with Poisson family error as richness values are count data) that analyze the raw plot values. The nested structure (plots within islands) can be included in the random structure of the model. This setting would further allow including information per plot (land cover class, habitat etc) in the analysis, if wanted. It would also allow differentiating an analysis on richness per plot with one on richness per island. It is possible that richness per plot is differently affected by land use than richness per island.”
→ We followed the reviewer’s suggestion (see also response to reviewer #1, comment 3) and fitted a nested Generalized Linear Mixed Effect Model on the plot-wise invertebrate biodiversity data (see response to reviewer #1 comment 3).
We obtained Remote Sensing Data for each island but not for each plot, thus including the information of an island’s relative amount of tree coverage is not possible to fit on the plot data (see also response to reviewer #2 comment 2 below).
2. “I would recommend analyzing differences in similarity in species composition using ordination (e.g., NMDS or CA) on the plot/species data. Possible predictor variables can be fitted with a post hoc fit or directly included in the analysis (CCA). This would allow a visualization of the difference in species composition with land-use types.”

- We followed the reviewer's suggestion and calculated a principal component analysis to better visualize the differences in the remote sensing data between island types for each island. Fitting the remote sensing data obtained for each island with the plot data is, however, not possible (see also comment above). Remote sensing parameters like %shrub coverage are measured island-wise using 1m² spatial resolution. Thus, a given plot (dimension 1x1 m) would have either grass, tree, or shrub coverage, rendering these as categorical variables when combined with the plot data. As CCA requires continuously measured variables for both parameter sets (in the case of our study: diversity and environmental parameters), it is thus not possible to fit the plot-wise diversity data with the island-wise remote sensing data, as suggested by reviewer #2.
3. "Islands are special and relevant regarding global biodiversity. They harbor less species per area, when compared to mainland settings, but an above average share of unique, often endemic species. Many of the species driven to extinction by humans have only occurred on islands. This context information and the island setting of this study should be placed in the introduction which currently builds the impression that the chosen islands are a random and representative case study. More background information on the biogeographic setting and land-use context of the islands would be helpful."
- We follow the reviewer's suggestion and fully revised the first paragraphs of the introduction to direct the storyline towards the biogeographic properties and unique characteristics of island ecosystems and their contribution to global biodiversity (see track changes).
4. "Title: I would suggest to remove 'permanent' and 'major'. Both words are not needed for the message."
- We followed the reviewer's suggestion and changed the title of the manuscript accordingly:
"Tourism and urban development as drivers for invertebrate diversity loss on tropical island"
5. "Line 16: Please remove 'pristine'. These ecosystems have been altered by humans even without permanent settlements."
- We followed the reviewer's suggestion and removed the word "pristine" at the given line and throughout the manuscript's text.
6. "Line 19: 'virtually absent' -> understudied?"
- In the given line, 'understudied' would not be the correct word. We therefore decided to keep the original wording.
7. "Line 17-19: Split in two sentences to improve readability?"
- We have completely revised the introductory paragraphs of our manuscript to implement the other suggestions of both reviewers to frame the manuscript more towards the unique biogeographic properties of islands.
8. "Lines 15-21: These sentences are too general for the setting of the case study. You could/should mention that this study is about islands, which are special and above average in their importance for biodiversity."
- We followed the reviewer's suggestion and included more information on unique properties and characteristics of islands at the given lines.

9. "Line 20: is 'urban' the right word for those small settlements?"
- ➔ The settlements on the investigated islands might appear small in their spatial extension, but when putting in relation to their total population, these islands have population densities of up > 10000 inhabitants/km² (e.g., Naifaru island 5800 inhabitants on 0.57 km² of island area), which is more densely inhabited than most European cities (e.g., Berlin 4200 inhabitants per km²). Referring to them as urban areas / islands should thus be an accurate description. We have included this in the revised version of the methods section in our manuscript together with a reference that summarises the definitions of 'urban areas' in different contexts:
"The inhabited islands had total human populations of ca. 800-5000. Due to their small total area, this results in extremely high population densities (3000 – 8000 inhabitants/km²), comparable or even exceeding those of many metropolitan continental urban areas. Therefore, these inhabited islands meet the criteria of most common definitions of urban areas and were referred to as 'urban islands' [26]."
10. "Line 21: The location of the case study should be included/mentioned here."
- ➔ We followed the reviewer's suggestion and included information on the location of the studied system at the given line.
11. "Line 48: You should mention the importance of islands for global diversity and their specialists (evolutionary history) here or in lines 77."
- ➔ We followed the reviewer's suggestion and included this information at the given line.
12. "Line 78: is 'urban' the right word for those small settlements?"
- ➔ See response to comment #9 above.
13. "Line 83: How do you know that the biogeographic history is the same?"
- ➔ The geological and biogeographic history of the Maldives is relatively well understood. The atoll islands are of volcanic origin, which formed roughly 55 million years ago. After being fully submerged during the last glacial period, the islands as they exist in their current started forming around 3000 years ago. They are oceanic islands that were never connected to any continental land mass and thus colonized primarily from the Indian subcontinent, in addition to a large set of introduced species that accumulated over the centuries of Maldives colonial history. Nevertheless, we decided to remove this information from the revised version of the manuscript.
14. "Lines 86-91: this are two predictions derived from the hypothesis that human land use causes the local loss of native invertebrates. Reformulate. How are non-natives treated in the study?"
- ➔ We did not differentiate between native and non-native species. Due to the young age of the Maldives (see comment above) and their volcanic origin, all species have been introduced to the islands at some point, either via natural processes or by humans, which makes it difficult to define what is native and non-native. At the same time, no endemic terrestrial invertebrate taxa are known from the Maldives, which can be clearly categorized as 'native'. Especially for invertebrates, which were at focus in this study, it is extremely difficult to trace back whether they have colonized the islands via natural processes or have been intentionally or unintentionally introduced by humans. To unravel their origin, detailed microsatellite-based population genetic studies on each species would be necessary, which are simply not available to this date. Thus, we cannot

differentiate between native and non-native invertebrate species and did not include this information in the statistical analysis of our study.

15. "Lines 112-114: You obviously had the expectation that those affect your dependent variable. Why was this not tested and included in the study?"
 - ➔ In line 112-114, we explain how ghost crab abundance was counted. The reviewer's comment likely refers to a different line in the method section, but without more context, we unfortunately cannot retrace which line this comment refers to and thus cannot address the reviewer's comment.

16. "Lines 156: See my general comment to the statistical analysis. Map of the study area with two different spatial scales (global context; regional setting)."
 - ➔ See comment above. We have revised the statistical analysis following the reviewer's suggestion to use a Generalized Linear Mixed Effect model.

17. "Figure 2: It is not clear to me if the boxplots represent the values per plot or per island. I would recommend showing the raw values per plot (1sqm). Please indicate significant differences as derived from the PostHoc test with letters. I would recommend to standardize the text size of all labels and reduce the white space by making the figure smaller."
 - ➔ We have completely revised figure 1 following both reviewers' suggestions.